# Effects of Safety Attitudes on Crossing Behaviours of Students Aged 10–18 Years: The Moderating Effects of Family Climate and Social Norms

**DOI:** 10.3390/bs15040415

**Published:** 2025-03-24

**Authors:** Qi Zhang, Shuo Yan, Long Sun

**Affiliations:** 1School of Education, Liaoning Normal University, Dalian116029, China; aqi7788@163.com; 2School of Psychology, Liaoning Normal University, Dalian 116029, China; yssiri@yeah.net

**Keywords:** crossing behaviour, safety attitudes, social norms, family climate, young pedestrian

## Abstract

This study focused on the effects of safety attitudes on young pedestrians’ risky and positive crossing behaviours, with an emphasis on the moderating role of social norms and the family climate. Four hundred young pedestrians aged 10~18 years agreed to participate in this study and were required to complete the survey, which included items related to risky and positive pedestrian crossing behaviours, social norms, safety attitudes and the family climate. Safety attitudes, social norms and the family climate had direct effects on pedestrians’ risky behaviours (aggressive, lapses and transgression), whereas only social norms could predict positive behaviours. Social norms and the family climate moderated the relationships between safety attitudes and transgressions, lapses and aggressive behaviour separately. More importantly, a three-way interaction was found, which indicated that social norms moderate the relationship between safety attitudes and transgression behaviours when the family climate is low. However, if parents actively monitor their offspring’s behaviour and act as positive role models, a stronger rule violation attitude does not increase their transgression behaviour under low risk-supportive peer norms. The findings suggest that family climate and social norms are important determinants of pedestrian crossing behaviour through interactions with safe attitudes, providing a theoretical framework for the development of safety interventions for pedestrians aged 10–18 years.

## 1. Introduction

In 2019, 2593 fatalities and 19619 injuries of pedestrians under the age of 15 years were recorded across the country by the National Bureau of Statistics of China ([30]). Fatal collisions and injuries in adolescents are frequent because their cognitive and perceptive skills have not reached their full development ([1]). Adolescents, characterized by limited cognitive abilities, high impulsivity and inaccurate assessments of danger, are more likely to engage in risky behaviours such as running red lights and abruptly accelerating across roads (e.g., [3]; [29]). To reduce the occurrence of such traffic accidents among young pedestrians, many studies have examined the characteristics of pedestrians’ crossing behaviours and their influencing factors (e.g., [23]; [29]; [32]). Although many studies have demonstrated that safety attitudes and peer and parental influences have important impacts on the safety of pedestrian behaviours (e.g., [13]; [20]; [24]), few studies have examined the interactions between safety attitudes, social norms and the family climate on the crossing behaviour of young pedestrians aged 10–18 years.

### 1.1. Pedestrian Behaviour

Typically, self-reported risk behaviours as a pedestrian are commonly examined in the literature. In this context, several self-reported instruments have been developed. [17] ([17]) developed the Pedestrian Behaviour Questionnaire (PBQ) in Chile. The PBQ contains 26 items divided into four factors: violation, error, lapses, and aggressive behaviour. Studies using the PBQ have revealed that male pedestrians commit more violations than female pedestrians do ([17]), and more errors are observed among young pedestrians (18–24 years) than among older pedestrians ([13]).

[9] ([9]) constructed the Pedestrian Behaviour Scale (PBS). The PBS has 32 items and is divided into 4 factors. Transgression behaviour is defined as making decisions that put pedestrians at risk, whether they violate the law or not. Lapses are defined as maladaptive behaviours associated with a lack of focus on a task. Positive behaviours are defined as those that soothe social interactions. Aggressive behaviour is defined as conflicting behaviour with other road users. The PBS by [9] ([9]) focuses on changes in the street crossing behaviour of pedestrians with ages ranging from 15 years to 78 years. The PBS was later validated in other countries ([27]), such as Serbia ([2]) and the USA ([5]). [9] ([9]) reported that individuals whose age is less than 35 years tend to commit more transgressions, offences and errors than those whose age is greater than 45 years. However, whether the original PBS can be used for younger people (aged 10–18 years) still needs more investigation.

Risk-taking behaviours during adolescence are well known, and numerous studies have focused on young teens’ risky behaviours as pedestrians and/or driver licence candidates ([3]; [7]; [23]; [32]). [23] ([23]) examined the relationships between crossing behaviours during early adolescence (ages 10~14) and risk perception and reported that risky behaviours while travelling as a pedestrian were more widespread among ninth-grade students and those who were more independent. Similarly, [13] ([13]) reported that younger people aged 18–24 years reported a greater risk of pedestrian behaviour than older adults aged over 34 years. To our knowledge, few studies have examined the characteristics of the crossing behaviour of young pedestrians aged 10–18 years and their influencing factors in China.

### 1.2. Theoretical Framework

It is widely believed that the formation of all behaviour is a process wherein external factors interact with personally held dispositions ([10]). Following this theoretical logic, several studies have examined the interactions of personal variables (internal factors) and situational variables (external factors) in the field of driving behaviour and reported that personal and external variables jointly affect driving behaviour (e.g., [4]; [31]). For instance, [4] ([4]) explored the moderating role of attitudes in the relationship between a family safety climate (i.e., noncommitment) and risky driving behaviours and reported that greater noncommitment was associated with more risky driving when young drivers had risk-supportive attitudes.

The present study adopted the same logic as [4] ([4]) and [31] ([31]), and we focused on safety attitudes and their interaction with social norms and the family climate on young pedestrians’ risky and positive behaviours. Attitudes are tendencies to evaluate an entity with some degree of favour or disfavour, ordinarily expressed in cognitive, affective and behavioural responses ([6]). Numerous studies have shown that a risk-supportive attitude results in more risky driving behaviour ([4]; [12]) or unsafe crossing behaviour ([8]; [13]; [15]; [32]). For example, [32] ([32]) reported that students aged 12–19 years have a more positive attitude towards crossing behaviour, experience more social approval from significant others and consider this behaviour to be more consistent with moral norms. Although some studies have shown that safety attitudes significantly affect individuals’ willingness and intention to cross the street safely ([13]; [15]), other studies have reported that safety attitudes are poor predictors of safety behaviours in countries that are prominently ‘collectivistic’ ([18]). This inconsistency suggests that the effect of safety attitudes on pedestrian crossing behaviour still needs more investigation.

With respect to the external variables, we use social norms and family climate as potential moderators of the effect of safety attitudes on pedestrians’ crossing behaviour. The family climate for road safety in this study refers to the perceptions of young teens regarding the values, perceptions, priorities and practices of their parents with respect to safe crossing. Social learning and socialization theories stress familial processes of behaviour transmission and focus more particularly on parents as offspring role models ([26]). For example, [26] ([26]) reported that a lower level of positive parental model and feedback leads to more reckless driving. Other studies have shown that young drivers who imitate their parents’ model for risky driving and parents tend not to invest time in safety education and ignore young drivers’ risky driving behaviour are associated with more self-reported reckless driving behaviours (e.g., [4]; [14]). In this context, this study takes the first step to explore parental modelling and positive feedback on their offspring’s risky and safe crossing behaviours.

Social norms can be seen as a person’s beliefs or understandings of a group about how members of this group should behave in certain situations. In this study, social norms refer to peer influence on young pedestrians. Studies have reported consistent results regarding the effects of social norms elicited from peers on young pedestrians’ crossing behaviours ([16]; [19]; [21]; [28]). For instance, [21] ([21]) reported that participants aged 16–18 years least often identified safe road-crossing sites when accompanied by a negative peer and more frequently identified dangerous road-crossing sites when accompanied by a positive peer. However, to our knowledge, the effects of social norms on young pedestrians’ crossing behaviour have largely not been investigated in China. Given that Chinese people tend to cross the street dangerously if enough people are waiting on the roadside, exploring whether the effects of social norms enhance or reduce young teens’ risky crossing behaviour can be highly valuable.

### 1.3. Objective and Hypotheses

The primary objective of the present study was to examine the effects of safety attitudes on pedestrian behaviour, and the moderating roles of social norms and the family climate in the relationships between safety attitudes and pedestrian behaviour were also examined. As portrayed in the hypothesized model (Figure 1), the independent variable is safety attitudes, the dependent variable is PBS factors and the moderating variables are social norms and family climate. Three hypotheses are formulated from these variables.

**Hypothesis** **1.**
*Safety attitudes affect both young pedestrians’ risky and positive behaviours.*


**Hypothesis** **2.**
*Social norms and the family climate directly influence pedestrian behaviour.*


**Hypothesis** **3.***Social norms and family climate moderate the effects of safety attitudes on pedestrians’ behaviour*.

## 2. Methods

### 2.1. Participants and Procedure

A total of 400 participants agreed to participate in this study. Participants were recruited from Dalian, Zhangzhou and Chengdu. After the participants were informed of the purpose of the study, 16 participants were unwilling to participate, and the remaining 384 signed a consent form and finished the paper and pen survey. The survey included items related to pedestrian behaviour, social norms, the family climate and safety attitudes. Participants first completed a self-reported demographic questionnaire in which they provided information such as sex, age, class, walking frequency and weekly walking time. Participants were then required to finish the survey within 30 min. Data were collected from 1 to 20 October 2024.

Finally, 370 valid data were obtained and 14 data were discarded because some answers were missing. Each participant received 5 RMB upon finishing the survey. The sample included 181 males (48.9%) and 189 females (51.1%). Participants ranged in age from 10–18 years (*M* = 13.99, *SD* = 2.53). Participants’ walking frequency ranged from 1 (less than once per day) to 4 (5~6 times per day) (*M* = 2.22, *SD* = 1.05), and their mean weekly walking time ranged from 1 (less than 150 min) to 3 (more than 300 min) (*M* = 2.06, *SD* = 0.81). Most of the participants (86.5%) were unaccompanied by adults, particularly when travelling to and from school as pedestrians or on public transport.

### 2.2. Measures

#### 2.2.1. PBS Scale

The original PBS contains 32 items. Participants were asked to rate the items on a 5-point Likert scale ranging from 1 (never) to 5 (always). The PBS has 4 factors: transgression (e.g., *I cross between vehicles stopped on the roadway in traffic jams*), lapses (e.g., *I become angry with another user and insult him*), positive behaviour (e.g., *I stop to let the pedestrians I meet by*) and aggressive behaviour (e.g., *I cross without looking, following other people who are crossing*). Given that the age of the participants in the original study ranged from 15–78 years, the factorial structure of the 32-item scale was first examined via exploratory factor analysis to assess its reliability in assessing the crossing behaviours of pedestrians aged 10–18 years.

#### 2.2.2. Social Norms

The social norm scale contains 6 items that address both the implicit and the explicit influences that respondents experience from their peers ([4]). The items were modified to better capture social norms among pedestrians. Both implicitly expressed peer norms (e.g., *my friend often uses his cell phone while crossing the street*) and explicitly expressed peer norms (e.g., *my friends will not mind if I do not look at the traffic lights occasionally*) were assessed. Participants were asked to rate the items on a 5-point Likert scale ranging from 1 (agree very much) to 5 (agree not at all). The total score was calculated by averaging the scores of the 6 items, with a higher score indicating a greater presence of risk-supportive peer norms. In this study, the internal consistency coefficient (Cronbach’s alpha) of this scale was 0.74.

#### 2.2.3. Family Climate

The family climate scale is derived from an existing scale, namely the Family Climate for Road Safety Scale (FCRSS) developed by [26] ([26]). This 6-item scale has been widely used in previous Chinese studies (e.g., [14]). Five items (e.g., *my parents are concerned about whether I cross the street safely*) from the FCRSS scale were used in this study, and all the items were modified to better capture the family climate perceived by young pedestrians. Participants were asked to rate the items on a 5-point Likert scale ranging from 1 (not at all) to 5 (very much). The total score was calculated by averaging the 5 items, with a higher score indicating a higher level of parental monitoring and modelling. In this study, the internal consistency coefficient (Cronbach’s alpha) for this scale is 0.85.

#### 2.2.4. Safety Attitudes

The safety attitudes scale was derived from an existing scale developed by [12] ([12]). The scale includes 6 items, which come from the rule violation factor of the original scale. The scale has been widely used in previous Chinese studies (e.g., [13]). The items are modified to better capture safety attitudes among young pedestrians (e.g., *to feel the destination in time, it is possible to occasionally break traffic rules*). Participants were asked to rate the items on a 5-point Likert scale ranging from 1 (strongly disagree) to 5 (strongly agree). The total score was calculated by averaging the scores of the 6 items, with a higher score indicating a higher level of rule violation. In this study, the internal consistency coefficient (Cronbach’s alpha) for this scale was 0.74.

### 2.3. Data Analysis

The data were analysed via SPSS 26.0. First, exploratory factor analysis (EFA) was conducted to test the factorial structure of the PBS in young pedestrians. Second, the correlations between social norms, the family climate, safety attitudes and the PBS factors were calculated. Third, the predictive role of social norms, the family climate and safety attitudes and their interaction with the PBS factors were analysed via hierarchical regression analysis. The interaction terms that were found to be significant in the regression analysis were further analysed via simple slope analysis to determine the nature of the interaction effects via the Hayes PROCESS macro Model 1 suggested by [11] ([11]).

### 2.4. Common Method Bias

Harman’s single-factor method was used to assess common method bias ([22]). The first factor accounted for 33.25% of the variance, and factors for which the eigenvalues exceeded 1.0 accounted for 58.86% of the variance. Thus, common method bias does not appear to be a serious problem in this study.

## 3. Results

### 3.1. Exploratory Factor Analysis

The original 32 items were subjected to an EFA via principal component analysis with oblique rotation. The sampling adequacy measure of Kaiser–Meyer–Olkin was 0.962, and Bartlett’s sphericity test was significant (chi-square = 7540.378, *p* < 0.001). The results revealed that four factors (eigenvalues > 1) explained 61.11% of the total variance. However, six items (e.g., *I realized I could not remember the route I just took*) that loaded on more than one factor were deleted.

The remaining 26 items are submitted to a second EFA again. The sampling adequacy measure of Kaiser–Meyer–Olkin was 0.951, and Bartlett’s sphericity test was significant (chi-square = 5619.987, *p* < 0.001). The results revealed that four factors (eigenvalues > 1) explained 61.85% of the total variance. Factor 1 (11 items), labelled transgression behaviour, explained 42.41% of the variance. Factor 2 (5 items), labelled positive behaviour, explained 9.82% of the variance. Factor 3 (5 items), labelled aggressive behaviour, explained 5.24% of the variance. Factor 4 (5 items), labelled lapses, explained 4.38% of the variance. The factor loadings are shown in Table 1. The summary statistics of the PBS factors are shown in Table 2. The factorial structure was comparable to that obtained in the original PBS ([9]).

### 3.2. Correlation Analysis

The correlations between the PBS factors and social norms, family climate and safety attitudes are shown in Table 3.

Table 3 shows that social norms and safety attitudes are positively correlated with transgression, lapses and aggressive behaviour and negatively correlated with positive behaviour. The family climate is negatively correlated with transgression, lapses and aggressive behaviour.

### 3.3. Hierarchical Regression Analyses

Several hierarchical regression analyses were conducted with transgression, lapses, and aggressive and positive behaviour as the dependent variables. In each regression, sociodemographic variables (e.g., sex, age and class) were entered in step 1. Social norms, safety attitudes and the family climate were entered in step 2. The interaction of the three independent variables was entered in step 3, and only the effects of those interactions that reached significance are shown in Table 4.

Table 3 shows that social norms, safety attitudes and family climate can significantly predict transgression, lapses and aggressive behaviour and explain 54.2%, 44.3% and 49.3% of the variance, respectively, whereas only social norms can predict positive behaviour and explain 2.6% of the variance. The interactions between social norms and safety attitudes, between family climate and safety attitudes and the three variables contributed to 4.6% of the variance in transgression behaviours. The interactions between social norms and safety attitudes and between family climate and safety attitudes contributed to 2.8% of the variance in lapses and 2.3% of the variance in aggressive behaviour.

### 3.4. Simple Slope Analyses

Significant interactions were further examined via the Hayes PROCESS macro-Model 1 suggested by [11] ([11]). Figure 2a shows that social norms moderated the relationship between safety attitudes and transgression behaviour. For high social norms (*b* = 0.59, *t* = 9.53, *p* < 0.01), a higher score on safety attitudes was associated with more transgression behaviour. For low social norms (*b* = −0.02, *t* = −0.37, *p* = 0.71), a higher score for safety attitudes did not increase transgression behaviour. Figure 2b,c revealed similar results: for high social norms, a higher score on safety attitudes was associated with more lapse behaviour (*b* = 0.60, *t* = 8.97, *p* < 0.01) and aggressive behaviour (*b* = 0.58, *t* = 9.20, *p* < 0.01). For low social norms, a higher score on safety attitudes did not increase lapse behaviour (*b* = 0.09, *t* = 1.43, *p* = 0.15) or aggressive behaviour (*b* = 0.02, *t* = 0.33, *p* = 0.74).

Figure 2d shows that the family climate moderated the relationship between safety attitudes and transgression behaviour. For both a low family climate (*b* = 0.51, *t* = 13.58, *p* < 0.01) and a high family climate (*b* = 0.15, *t* = 2.97, *p* < 0.05), a higher score on safety attitudes was associated with more transgression behaviour. Figure 2e shows that for both a low family climate (*b* = 0.55, *t* = 13.73, *p* < 0.01) and a high family climate (*b* = 0.20, *t* = 3.61, *p* < 0.01), a higher score on safety attitudes was associated with more aggressive behaviour. Similarly, Figure 2f shows that for both a low family climate (*b* = 0.63, *t* = 13.98, *p* < 0.01) and a high family climate (*b* = 0.23, *t* = 3.64, *p* < 0.01), a higher score on safety attitudes was associated with more lapse behaviour.

We conducted a simple slope analysis to examine the three-way interaction effects of social norms, safety attitudes and the family climate on transgression behaviours. Figure 2g shows that when social norms are high, a higher score on safety attitudes leads to more transgression behaviour when both family climate are high (*b* = 0.31, *t* = 3.83, *p* < 0.01) and lower (*b* = 0.42, *t* = 5.17, *p* < 0.01). Figure 2h shows that when social norms are low, a higher score on safety attitudes leads to more transgression behaviour when the family climate is low (*b* = 0.43, *t* = 4.88, *p* < 0.01) but not when the family climate is high (*b* = −0.05, *t* = −0.75, *p* = 0.45).

## 4. Discussion

This study took the first step in investigating the relationships between young pedestrians’ risky and positive crossing behaviours, safety attitudes, family climate and social norms. Our results revealed significant associations between the studied variables and the moderating role of family climate and social norms in relation to safety attitudes and crossing behaviours. The findings not only support the theoretical model of interactions between external factors and personal variables on crossing behaviours but also provide a theoretical framework for developing interventions to ensure the crossing safety of younger pedestrians.

Safety attitudes and social norms are important determinants of pedestrian crossing behaviour. This study also confirmed that perceived parental influence (family climate) played an important role in determining risky and positive crossing behaviour of young pedestrians. Our results show that safety attitudes, social norms and family climate can significantly predict transgression, lapses and aggressive behaviour, whereas only social norms and family climate can significantly predict positive behaviour, which supports research Hypotheses 1 and 2. The present study revealed that social norms moderate the effect of safety attitudes on three risky crossing behaviours (i.e., transgression, lapses and aggressive behaviour). Only when the level of risk-supportive peer norms is high (social norm) is a stronger rule violation attitude associated with more transgression, lapses and aggressive behaviour. These results are consistent with the findings of several previous studies showing that a risk-supportive attitude and negative peer influence are associated with more risky crossing behaviours (e.g., [13]; [16]; [19]; [21]; [28]; [32]).

More importantly, family climate moderates the effect of safety attitudes on the three risky crossing behaviours (i.e., transgression, lapses and aggressive). A higher rule violation attitude leads to more transgression, lapses and aggressive behaviour, regardless of whether the family climate is high or low. Consistent findings in driving safety have demonstrated that a lack of parental commitment to road safety leads to an increase in risky behaviours among young drivers ([4]; [14]; [25]). These results indicated that positive communication and consensus between parents and offspring on road safety could help reduce risky crossing behaviours of young pedestrians aged 10–18 years.

The present study contributed to the literature in that we found significant interactions between safety attitudes, social norms and the family climate in terms of transgression behaviour. Our results show that regardless of whether the level of risk-supportive peer norms is low or high (social norms), a higher rule violation attitude results in more transgression behaviours if parents do not give much attention to monitoring their offspring’s crossing behaviour on the road or setting a model for their offspring by obeying traffic rules and crossing the street safely. However, a higher score for family climate did not increase transgression behaviours as much as a lower score for family climate. These findings suggest that a higher score for family climate can serve as a protective factor in reducing transgression behaviour. This finding was further confirmed by our results, which indicated that a stronger rule violation attitude did not increase their transgression behaviours under low risk-supportive peer norms. Even though young pedestrians might confront the negative influence of their friends ([16]; [19]; [28]), parents can still minimize these adverse effects by teaching their children to obey traffic rules and set examples in safe crossing behaviour.

## 5. Implications and Limitations

These research results have certain practical applications. First, the relationships among social norms, safety attitudes, the family climate and various crossing behaviours provide a theoretical approach for better understanding both risky and positive crossing behaviours of young pedestrians aged 10–18 years. Second, our findings highlight that safety education programs should be tailored to address both individual attitudes and peer influences. Given that a low score on social norms does not increase transgression, lapses or aggressive behaviours, it might be of great value in cultivating a shared commitment to safe crossing among young pedestrians. For policy makers and practitioners, safety education and interventions that jointly include young pedestrians and their peers might be more effective. Third, the present study contributes to our understanding of how familial, peer and individual factors interact to influence risky crossing behaviours. Our findings suggest that school and community-based initiatives that engage both young pedestrians and their parents could be more effective in reinforcing safe crossing behaviours. For instance, education and counselling should be provided to young pedestrians and their peers to change their safety attitudes, whereas guidance should be provided to parents to help foster a positive family climate for road safety.

This study has several limitations. The first limitation is that the study is based on a small sample, and the sample might not represent the whole population in China. Future studies with larger samples should be conducted to further examine the generalizability of the results obtained in this study. Although we recruited participants from three cities in China, the differences in the environmental factors (e.g., speed limit, road type, population density and weather conditions) of the three cities could affect pedestrians’ crossing behaviours. Another limitation is that, as in most previous studies ([4]; [31]), the family climate was used as a moderator. In this study, the items related to family climate were limited to parental monitoring and modelling. Given that the family climate for road safety is a multidimensional structure, future studies are recommended to explore the effects of other factors of the family climate on pedestrians’ crossing behaviour. A third limitation is that the relationships among the studied variables may be inaccurate due to social desirability. In the future, more objective indicators of crossing behaviours should be included.

## 6. Conclusions

This study examined the relationships among the safety attitudes, family climate, social norms and crossing behaviours of pedestrians aged 10–18 years. Our results showed that social norms, family climate and safety attitudes can significantly predict risky crossing behaviours (transgression, lapses and aggressive behaviour). More importantly, we found that the effects of safety attitudes on risky crossing behaviours were moderated by social norms and the family climate. The findings may provide a theoretical framework to understand young pedestrians’ behaviour and assist in developing safety education and interventions. However, the framework itself cannot directly reduce the number of risk-taking children. Instead, it offers valuable insights for creating more comprehensive safety strategies.

## Figures and Tables

**Figure 1 behavsci-15-00415-f001:**
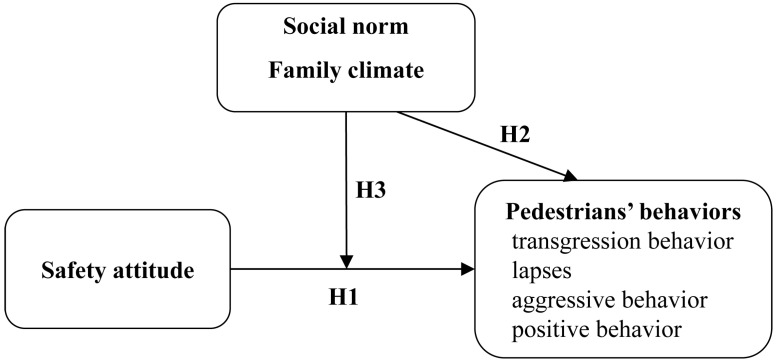
Hypothesized model of this study.

**Figure 2 behavsci-15-00415-f002:**
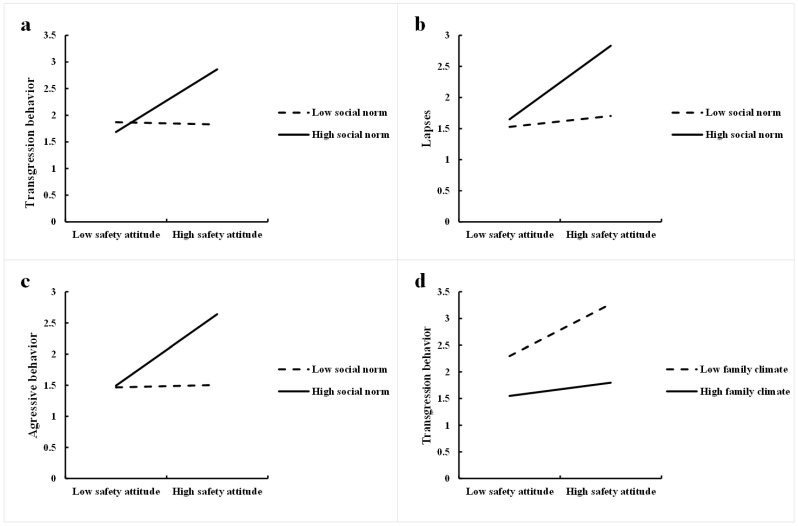
(**a**) Interaction between safety attitudes and social norms predicts transgression behaviour. (**b**) Interaction between safety attitudes and social norms predicts lapse behaviour. (**c**) Interaction between safety attitudes and social norms predicts aggressive behaviour. (**d**) Interaction between safety attitudes and the family climate predicts transgression behaviour. (**e**) Interaction between safety attitudes and the family climate predicts aggressive behaviour. (**f**) Interaction between safety attitudes and the family climate predicts lapse behaviour. (**g**) Interaction between safety attitudes and the family climate predicts transgression behaviour (SN + 1SD). (**h**) Interaction between safety attitudes and the family climate predicts transgression behaviour (SN − 1SD).

**Table 1 behavsci-15-00415-t001:** Factor loadings of the PBS.

Items (How Often Do You…)	Factor 1	Factor 2	Factor 3	Factor 4
1. I cross the street between parked cars	0.82			
2. I cross between vehicles stopped on the roadway in traffic jams	0.81			
3. I cross diagonally to save time	0.73			
4. I start walking across the street, but I have to run the rest of the way to avoid oncoming vehicles	0.69			
5. I start to cross on a pedestrian crossing and I finish crossing diagonally to save time	0.64			
6. I walk on the roadway to be next to my friends on the sidewalk or to overtake someone who is walking slower than I am	0.63			
7. I cross even though the traffic light is still green for vehicles	0.62			
8. I cross outside the pedestrian crossing even if there is one less than 50 m away	0.61			
9. I cross the street even though the pedestrian light is red	0.57			
10. I look at the traffic light and start crossing as soon as it turns red	0.56			
11. I cross while talking on my cell phone or listening to music on my headphones	0.49			
12. I walk on the right-hand side of the sidewalk so as not to bother the pedestrians I meet		0.75		
13. When I am accompanied by other pedestrians, I walk in single file on narrow sidewalks so as not to bother the pedestrians I meet		0.74		
14. I stop to let the pedestrians I meet by		0.72		
15. I let a car go by, even if I have the right-of-way, if there is no other vehicle behind it		0.58		
16. I thank a driver who stops to let me cross		0.46		
17. I get angry with a driver and hit his vehicle			0.74	
18. I cross very slowly to annoy a driver			0.71	
19. I get angry with another user (pedestrian, driver, cyclist, etc.) and I yell at him			0.70	
20. I get angry with another user and insult him			0.69	
21. I get angry with another user (pedestrian, driver, etc.) and I make a hand gesture			0.69	
22. I forget to look before crossing because I want to join someone on the sidewalk on the other side				−0.87
23. I cross without looking because I am talking with someone				−0.80
24. I forget to look before crossing because I am thinking about something else				−0.75
25. I realize that I have crossed several streets and intersections without paying attention to traffic				−0.75
26. I cross without looking, following other people who are crossing				−0.61

**Table 2 behavsci-15-00415-t002:** Summary statistics of PBS factors.

PBS Factors	Number of Item	Cronbach’s Alpha	Cronbach’s Alpha ^a^	*M*	*Sd*	Kurtosis	Skewness
Transgression	11	0.92	0.89	2.25	0.96	0.27	1.11
Positive behaviour	5	0.69	0.53	2.93	0.96	−0.80	0.11
Aggressive behaviour	5	0.91	0.70	1.94	1.00	−0.85	1.29
Lapses	5	0.90	0.83	2.08	1.06	0.48	1.19

Note: ^a^ Reliability of the original PBS ([9]).

**Table 3 behavsci-15-00415-t003:** Correlations between PBS factors, social norms, family climate and safety attitudes.

Variables	Transgression	Lapses	Aggressive Behaviour	Positive Behaviour
Lapses	0.71 **			
Aggressive behaviour	0.69 **	0.71 **		
Positive behaviour	−0.06	−0.20 **	−0.24 **	
Social norm	0.47 **	0.54 **	0.53 **	−0.19 **
Family climate	−0.62 **	−0.44 **	−0.54 **	0.10
Safety attitude	0.45 **	0.52 **	0.49 **	−0.15 **

Note. ** *p* < 0.01.

**Table 4 behavsci-15-00415-t004:** Results of hierarchical regression analysis.

Variable	Transgression			Lapses			Aggressive Behaviour			Positive Behaviour		
	*B*	*t*	*p*	*B*	*t*	*p*	*B*	*t*	*p*	*B*	*t*	*p*
Step 1												
Sex	0.02	0.55	0.581	−0.04	−1.12	0.265	−0.37	0.71	0.712	0.06	1.20	0.233
Age	0.19	0.80	0.428	0.14	0.53	0.598	1.63	0.11	0.105	−0.52	−1.47	0.142
Class	−0.17	−0.72	0.472	−0.11	−0.40	0.692	−1.25	0.21	0.214	0.40	1.13	0.259
Walking frequency	0.02	0.45	0.654	−0.02	−0.62	0.533	−0.65	0.52	0.519	−0.11	−2.20	0.028
Walking time	−0.04	−1.17	0.243	0.02	0.61	0.543	−0.02	0.99	0.988	−0.11	−2.14	0.033
Adjusted *R*^2^	0.017 *			0.028 **			0.047 **			0.042 **		
Step 2												
Social norm	0.12	2.76	0.006	0.22	4.24	0.001	0.20	4.29	0.001	−0.139	−2.13	0.034
Safety attitude	0.29	6.88	0.001	0.30	6.26	0.001	0.27	5.94	0.001	−0.024	−0.38	0.708
Family climate	−0.63	−11.79	0.001	−0.37	−6.16	0.001	−0.44	−7.90	0.001	0.09	1.65	0.10
Adjusted *R*^2^	0.542 **			0.443 **			0.493 **			0.026 **		
Step 3												
SA × SN	0.12	2.80	0.001	0.11	2.47	0.014	0.12	2.82	0.005	-	-	
SA × FC	−0.19	−4.88	0.001	−0.15	−3.32	0.001	−0.12	−2.86	0.004	-	-	
SN × SA × FC	0.176	3.30	0.001	−	-		-	-		-	-	
Adjusted *R*^2^	0.046 **			0.028 **			0.023 **					
Total *R*^2^	0.605 **			0.499 **			0.563 **			0.068 **		

Note: SN = social norm, SA = safety attitude, FC = family climate, * *p* < 0.05; ** *p* < 0.01.

## Data Availability

The original data presented in the study are openly available in [FigShare] at [10.6084/m9.figshare.28340681].

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
