# Peer review of "Effects of Safety Attitudes on Crossing Behaviours of Students Aged 10–18 Years: The Moderating Effects of Family Climate and Social Norms"

_behavsci, 2025, doi:10.3390/bs15040415_

Round 1
Reviewer 1 Report
Comments and Suggestions for Authors
General comments
- The research provides useful insights and adds to existing knowledge. However, some issues need to be addressed. Specifically, there are concerns about the transparency of the items and the appropriate use of terminology in the study.
Introduction/ literature review
- The paper could benefit from restructuring its introduction. It would be helpful to explain upfront why the focus group are young pedestrians, why specific factors are being examined, and how the study's findings are useful beyond academic circles.
- The study mentions overall traffic accidents and then states, “To reduce the occurrence of such traffic accidents among pedestrians” without providing statistics on pedestrian accidents or information about the focus group. Consider revising this section (Line 35-38).
- There is several literature examining risky behaviour among younger pedestrians aged 10-18 years, including crossing behaviour. Additionally, the effects of social norms on pedestrian crossing behaviour have been investigated internationally (see section 1.2). Please revisit previous literature.
Theoretical framework/ Methods
- The paper should include questions (items) in this study to make it easier for readers to follow. More importantly, this will help justify and validate whether the questions truly reflect the behaviour in question, especially since the paper mentions that the questions were modified from previous studies.
- The study claims that 400 samples from three cities geographically represent China (section 2.1), but the limitation section states that the sample might not represent the whole population. While 400 samples may not represent the entire country, it is still valuable for piloting other studies. Consider removing the word “geographically represent” in section 2.1 or statistically validate the sample size through a stratified sampling process to ensure representativeness and precision.
- The study used samples from three cities, but it is unclear if these cities share the same road rules or environments. Was the participants' general information collected? This impacts their behaviour towards road safety and should be explained or included in the limitations.
- Social norms encompass peers, acquaintances, unacquainted individuals, family members, and the general public. If the research focuses on peer pressures, this should be clearly stated, and the term 'social norms' may not be applicable. Additional literature on peer pressure could be included.
- Similarly, “Family climate” includes relationships, emotional bonds and the level of support and conflict within a family etc. The study focuses on parents’ perceptions, attitudes, and behaviours, which are only part of the whole picture. Although this limitation is acknowledged, it would be helpful to explain what “family climate” entails before discussing the specific aspects considered (line 110). This also raises the question of whether the term “family climate” is the right term to use given the specific aspects being studied.
- The study refers to the family climate for road safety as the perceptions of young teens regarding their parents' or family's values, perceptions, priorities, and practices related to safe driving (line 110). The paper needs to explain and validate how these perceptions of safe driving relate to safe crossing, even though they are understood to be potentially related.
- How do authors address the potential issues with self-report surveys, such as participants providing socially desirable answers and the possibility of varied interpretations of the questions? If these are considered study limitations, it would be beneficial to clearly explain them.
Discussion
- The study should focus on highlighting its own findings rather than repeatedly mentioning previous studies. While it is important to explain how these findings align with previous research, the key is to emphasise your study's unique contributions.
- Please review section 4.2. Ensure that the driving behaviours from previous studies are clearly connected to this study to make the results more meaningful.
Implications/ Limitations/ Conclusion
- It would be beneficial if the author could elaborate on the significance of these findings and their potential impact on future research, policy, practice, or theory. While it is valuable to reference implications from previous studies, sharing the author's unique insights would provide more value to the audience.
- Consider revising this “The findings not only provide a theoretical framework to reduce younger pedestrians’ risky crossing behaviour but are also valuable for developing safety education and interventions from a more comprehensive perspective of family, peers and younger pedestrians themselves.” The findings may provide a theoretical framework to understand younger pedestrians’ behaviour and assist in developing safety education and interventions. However, the framework itself cannot directly reduce the number of risk-taking children. Instead, it offers valuable insights for creating more comprehensive safety strategies.
Author Response
The paper could benefit from restructuring its introduction. It would be helpful to explain upfront why the focus group are young pedestrians, why specific factors are being examined, and how the study's findings are useful beyond academic circles.
Response: As the reviewer suggested, the first paragrahp of introdcution was revised. Please see page 1-2. Or you can see the revision below.
In 2019, 2593 fatalities and 19619 injuries of pedestrians under the age of 15 years were recorded across the country by the National Bureau of Statistics of China (Xinhua, 2021). Fatal collisions and injuries in adolescents are frequent because their cognitive and perceptive skills have not reached their full development (Adminaite et al., 2018). Adolescents, characterized by limited cognitive abilities, high impulsivity, and inaccurate assessments of danger, are more likely to engage in risky behaviors such as running red lights and abruptly accelerating across roads (e.g., Biassoni et al., 2018; Wang et al., 2022). To reduce the occurrence of such traffic accidents among young pedestrians, many studies have examined the characteristics of pedestrians’ crossing behaviours and their influencing factors (e.g., Salducco et al., 2022; Wang et al., 2022; Zhou & Horrey, 2010). Although many studies have demonstrated that safety attitudes and peer and parental influences have important impacts on the safety of pedestrain behaviours (e.g., Liu et al., 2021; Papadimitriou et al., 2013; Soole et al., 2011), few studies have examined the interactions between safety attitudes, social norms and the family climate on the crossing behaviour of young pedestrians aged 10~18 years.
The study mentions overall traffic accidents and then states, “To reduce the occurrence of such traffic accidents among pedestrians” without providing statistics on pedestrian accidents or information about the focus group. Consider revising this section (Line 35-38).
Response: As the reviewer suggested, the first paragrahp of introdcution was revised. Please see our response for the former commnent.
There is several literature examining risky behaviour among younger pedestrians aged 10-18 years, including crossing behaviour. Additionally, the effects of social norms on pedestrian crossing behaviour have been investigated internationally (see section 1.2). Please revisit previous literature.
Response: As the reviewer suggested, previous studies concerning social norms and pedestrian crossing behaviours were revisited and cited. Please see page 3. Or you can see the revision below.
Social norms can be seen as a person’s beliefs or understandings of a group about how members of this group should behave in certain situations. In this study, social norms refers to peer influence on young pedestrian. Studies have reported consistent results regarding the effects of social norms elicited from peers on young pedestrians’ crossing behaviours (Morrongiello et al., 2019; O'Neal et al., 2019; Pfeffer & Hunter, 2013; Wang et al., 2024). For instance, Pfeffer and Hunter (2013) reported that participants aged 16-18 years least often identified safe road-crossing sites when accompanied by a negative peer and more frequently identified dangerous road-crossing sites when accompanied by a positive peer...
Theoretical framework/ Methods
The paper should include questions (items) in this study to make it easier for readers to follow. More importantly, this will help justify and validate whether the questions truly reflect the behaviour in question, especially since the paper mentions that the questions were modified from previous studies.
Response: As the reviewer suggested, one item for each factor of the scales used in this study was added. Please see page 5. Or you can see the example of revision below.
2.2.2. Social norms
The social norm scale contains 6 items that address both the implicit and the explicit influences that respondents experience from their peers (Carpentier et al., 2014). The items were modified to better capture social norms among pedestrians. Both implicitly expressed peer norms (e.g., my friend often uses his cell phone while crossing the street) and explicitly expressed peer norms (e.g., my friends will not mind if I do not look at the traffic lights occasionally) were assessed. Participants were asked ...
The study claims that 400 samples from three cities geographically represent China (section 2.1), but the limitation section states that the sample might not represent the whole population. While 400 samples may not represent the entire country, it is still valuable for piloting other studies. Consider removing the word “geographically represent” in section 2.1 or statistically validate the sample size through a stratified sampling process to ensure representativeness and precision.
Response: As the reviewer suggested, the word “geographically represent” in section 2.1 was removed. Please see page 4. Or you can see the example of revision below.
2.1. Participants and procedure
A total of 400 participants agreed to participate in this study. Participants were recruited from Dalian, Zhangzhou, and Chengdu. After the participants were informed of the purpose of the study, 16 participants were unwilling to participate, ...
The study used samples from three cities, but it is unclear if these cities share the same road rules or environments. Was the participants' general information collected? This impacts their behaviour towards road safety and should be explained or included in the limitations.
Response: The general road rules are the same in the three cities, while the infrastructure and traffic environment in the three cities might be different. As the reviewer suggested, this was mentioned as a limitation in this study. Please see page 12. Or you can see the revision below.
This study has several limitations. The first limitation is that the study is based on a small sample, and the sample might not represent the whole population in China. Future studies with larger samples should be conducted to further examine the generalizability of the results obtained in this study. Although we recruited participants from three cities in China, the differences in the environmental factors (e.g., speed limit, road type, population density, weather conditions) of the three cities could affect pedestrians’ crossing behaviours...
Social norms encompass peers, acquaintances, unacquainted individuals, family members, and the general public. If the research focuses on peer pressures, this should be clearly stated, and the term 'social norms' may not be applicable. Additional literature on peer pressure could be included.
Response: Thanks very much for your valuable suggestions. As the reviewer suggested, this section was revised to clearly state social norms refers to peer influence in this study. Please see page 3. Or you can see the revision below.
Social norms can be seen as a person’s beliefs or understandings of a group about how members of this group should behave in certain situations. In this study, social norms refers to peer influence on young pedestrian. Studies have reported consistent results regarding the effects of social norms elicited from peers on young pedestrians’ crossing behaviours (Morrongiello et al., 2019; O'Neal et al., 2019; Pfeffer & Hunter, 2013; Wang et al., 2024)...
Similarly, “Family climate” includes relationships, emotional bonds and the level of support and conflict within a family etc. The study focuses on parents’ perceptions, attitudes, and behaviours, which are only part of the whole picture. Although this limitation is acknowledged, it would be helpful to explain what “family climate” entails before discussing the specific aspects considered (line 110). This also raises the question of whether the term “family climate” is the right term to use given the specific aspects being studied.
Response: Thanks very much for your valuable suggestions. In this study, family climate refers to only parental influence. This sentence was revised to make it clear. The other aspects of family climate was not the focus of the present study and we have already mentioned this as a limitation. Please see page 3. Or you can see the revision below.
With respect to the external variables, we use social norms and family climate as potential moderators of the effect of safety attitudes on pedestrians’ crossing behaviour. The family climate for road safety in this study refers to the perceptions of young teens regarding the values, perceptions, priorities and practices of their parents with respect to safe crossing...
The study refers to the family climate for road safety as the perceptions of young teens regarding their parents' or family's values, perceptions, priorities, and practices related to safe driving (line 110). The paper needs to explain and validate how these perceptions of safe driving relate to safe crossing, even though they are understood to be potentially related.
Response: Thanks very much for your valuable suggestions. In this study, family climate refers to only parental influence. This sentence was revised to make it clear. Please see our response for the former question.
How do authors address the potential issues with self-report surveys, such as participants providing socially desirable answers and the possibility of varied interpretations of the questions? If these are considered study limitations, it would be beneficial to clearly explain them.
Response: As the reviewer suggested, this was mentioned as a limitation in this study. Please see page 12. Or you can see the example of revision below.
This study has several limitations. … A third limitation is that the relationships among the studied variables may be inaccurate due to social desirability. In the future, more objective indicators of crossing behaviours should be included.
Discussion
The study should focus on highlighting its own findings rather than repeatedly mentioning previous studies. While it is important to explain how these findings align with previous research, the key is to emphasise your study's unique contributions.
Response: As the reviewer suggested, the discussion section were re-organized to emphasise the key findings of the present study. Please see page 11. Example of the revision could be seen below.
Safety attitudes and social norms are important determinants of pedestrian crossing behaviour. This study also confirmed that perceived parental influence (family climate) played an important role in determining risky and positive crossing behaviour of young pedestrians. Our results show that safety attitudes, social norms, and family climate can significantly predict transgression, lapses, and aggressive behaviour, whereas only social norms and family climate can significantly predict positive behaviour, which supports research Hypotheses 1 and 2. …These results are consistent with the findings of several previous studies showing that a risk-supportive attitude and negative peer influence are associated with more risky crossing behaviours (e.g., Liu et al., 2021; Morrongiello et al., 2019; O'Neal et al., 2019; Pfeffer & Hunter, 2013; Wang et al., 2024; Zhou & Horrey, 2010).
More importantly, family climate moderates the effect of safety attitudes on the three risky crossing behaviours (i.e., transgression, lapses and aggressive)... These results indicated that positive communication and consensus between parents and offspring on road safety could help reduce risky crossing behaviours of young pedestrians aged 10~18 years.
The present study contributed to the literature in that we found significant interactions between safety attitudes, social norms and the family climate in terms of transgression behaviour. Our results show that...
Please review section 4.2. Ensure that the driving behaviours from previous studies are clearly connected to this study to make the results more meaningful.
Response: As the reviewer suggested, section 4.2 was revised. The references related to driving behaivours were replaced by those related to crossing behaviours. Please see page 11. Or you can see the revision below.
…Only when the level of risk-supportive peer norms is high (social norm) is a stronger rule violation attitude associated with more transgression, lapses and aggressive behaviour. These results are consistent with the findings of several previous studies showing that a risk-supportive attitude and negative peer influence are associated with more risky crossing behaviours (e.g., Liu et al., 2021; Morrongiello et al., 2019; O'Neal et al., 2019; Pfeffer & Hunter, 2013; Wang et al., 2024; Zhou & Horrey, 2010).
Implications/ Limitations/ Conclusion
It would be beneficial if the author could elaborate on the significance of these findings and their potential impact on future research, policy, practice, or theory. While it is valuable to reference implications from previous studies, sharing the author's unique insights would provide more value to the audience.
Response: As the reviewer suggested, these sentences were revised. Please see page 12. Or you can see the revision below.
The research results have certain practical applications. First, the relationships among social norms, safety attitudes, the family climate and various crossing behaviours provide a theoretical approach for better understanding both risky and positive crossing behaviours of young pedestrians aged 10~18 years. Second, our findings highlight that safety education programs should be tailored to address both individual attitudes and peer influences. Given that a low score on social norms does not increase transgression, lapse or aggressive behaviours, it might be of great value in cultivating a shared commitment to safe crossing among young pedestrians. For policy-makers and practitioners, safety education and interventions that jointly include young pedestrians and their peers might be more effective. Third, the present study contributes to our understanding of how familial, peers’ and individual factors interact to influence risky crossing behaviours. Our findings suggest that school and community-based initiatives that engage both young pedestrians and their parents could be more effective in reinforcing safe crossing behaviours. For instance, education and counselling should be provided to young pedestrians and their peers to change their safety attitudes, whereas guidance should be provided to parents to help foster a positive family climate for road safety.
Consider revising this “The findings not only provide a theoretical framework to reduce younger pedestrians’ risky crossing behaviour but are also valuable for developing safety education and interventions from a more comprehensive perspective of family, peers and younger pedestrians themselves.” The findings may provide a theoretical framework to understand younger pedestrians’ behaviour and assist in developing safety education and interventions. However, the framework itself cannot directly reduce the number of risk-taking children. Instead, it offers valuable insights for creating more comprehensive safety strategies.
Response: As the reviewer suggested, these sentences were revised. Please see page 13. Or you can see the revision below.
- Conclusions
This study examined the relationships among the safety attitudes, family climate, social norms and crossing behaviours of pedestrians aged 10~18 years…The findings may provide a theoretical framework to understand young pedestrians’ behaviour and assist in developing safety education and interventions. However, the framework itself cannot directly reduce the number of risk-taking children. Instead, it offers valuable insights for creating more comprehensive safety strategies.

Reviewer 2 Report
Comments and Suggestions for Authors
Section 1.1: Please consider other studies that focus on PBS (e.g. https://doi.org/10.1016/j.aap.2021.106509; https://doi.org/10.1016/j.trf.2016.02.004) as there are some differences in items, age and gender that might help the authors to explain their results.
More information is needed on the model, including justification, as we could argue that social norms might also affect family climate, so authors might consider moderated moderation.
Please include a table of EFA results (i.e. factor loadings, etc.).
Please report exact p-values for each statistic (e.g. Table 3).
Please review some methodological limitations such as social desirability (e.g. https://doi.org/10.1016/j.trf.2021.11.009).
The manuscript is well written and considers variable factors. I would strongly encourage authors to also discuss traffic climate (e.g., https://doi.org/10.1016/j.aap.2018.01.031; https://doi.org/10.1016/j.trf.2025.01.030), as it may help explain other differences and cross-cultural variability for future studies.
Author Response
Section 1.1: Please consider other studies that focus on PBS (e.g. https://doi.org/10.1016/j.aap.2021.106509; https://doi.org/10.1016/j.trf.2016.02.004) as there are some differences in items, age and gender that might help the authors to explain their results.
Response: As the reviewer suggested, other studies that focus on PBS were added in this study. Please see page 2, 1.1. Pedestrian behaviour. Or you can see the revision below.
… The PBS by Granié et al. (2013) focuses on changes in the street crossing behaviour of pedestrians with ages ranging from 15 years to 78 years. The PBS was later validated in other counties (Vandroux et al., 2022), such as Serbia (Antić et al., 2016) and the USA (Deb et al., 2017)...
More information is needed on the model, including justification, as we could argue that social norms might also affect family climate, so authors might consider moderated moderation.
Response: Thanks very much for your valuable suggestion. We would like to explore the possible interactions between only external variables, such as social norms and family climate, in future studies. Hence, we would not examine the possible moderated moderation in this study to limit the over-length of the whole article.
Please include a table of EFA results (i.e. factor loadings, etc.).
Response: As the reviewer suggested, the results of EFA was added. Please see page 6-7, Table 1. Or you can see the revision below.
Table 1. Factor loadings of the PBS.
|
Items (how often do you…) |
Factor 1 |
Factor 2 |
Factor 3 |
Factor 4 |
|
1.I cross the street between parked cars |
0.82 |
|
|
|
|
2.I cross between vehicles stopped on the roadway in traffic jams |
0.81 |
|
|
|
|
3.I cross diagonally to save time |
0.73 |
|
|
|
|
4.I start walking across the street, but I have to run the rest of the way to avoid oncoming vehicles |
0.69 |
|
|
|
|
5.I start to cross on a pedestrian crossing and I finish crossing diagonally to save time |
0.64 |
|
|
|
|
6.I walk on the roadway to be next to my friends on the sidewalk or to overtake someone who is walking slower than I am |
0.63 |
|
|
|
|
7.I cross even though the traffic light is still green for vehicles |
0.62 |
|
|
|
|
8.I cross outside the pedestrian crossing even if there is one less than 50 m away |
0.61 |
|
|
|
|
9.I cross the street even though the pedestrian light is red |
0.57 |
|
|
|
|
10.I look at the traffic light and start crossing as soon as it turns red |
0.56 |
|
|
|
|
11.I cross while talking on my cell phone or listing to music on my headphones |
0.49 |
|
|
|
|
12.I walk on the right-hand side of the sidewalk so as not to bother the pedestrians I meet |
|
0.75 |
|
|
|
13.When I am accompanied by other pedestrians, I walk in single file on narrow sidewalks so as not to bother the pedestrians I meet |
|
0.74 |
|
|
|
14.I stop to let the pedestrians I meet by |
|
0.72 |
|
|
|
15.I let a car go by, even if I have the right-of-way, if there is no other vehicle behind it |
|
0.58 |
|
|
|
16.I thank a driver who stops to let me cross |
|
0.46 |
|
|
|
17.I get angry with a driver and hit his vehicle |
|
|
0.74 |
|
|
18.I cross very slowly to annoy a driver |
|
|
0.71 |
|
|
19.I get angry with another user (pedestrian, driver, cyclist, etc.) and I yell at him |
|
|
0.70 |
|
|
20.I get angry with another user and insult him |
|
|
0.69 |
|
|
21.I get angry with another user (pedestrian, driver, etc.) and I make a hand gesture |
|
|
0.69 |
|
|
22.I forget to look before crossing because I want to join someone on the sidewalk on the other side |
|
|
|
-0.87 |
|
23.I cross without looking because I am talking with someone |
|
|
|
-0.80 |
|
24.I forget to look before crossing because I am thinking about something else |
|
|
|
-0.75 |
|
25.I realize that I have crossed several streets and intersections without paying attention to traffic |
|
|
|
-0.75 |
|
26.I cross without looking, following other people who are crossing |
|
|
|
-0.61 |
Please report exact p-values for each statistic (e.g. Table 3).
Response: As the reviewer suggested, the exact p-values were reported. Please see page 8, Table 4. Or you can see the revision below.
Table 4. Results of hierarchical regression analysis.
|
Variable |
Transgression |
Lapses |
Aggressive behaviour |
Positive behaviour |
|||||||||
|
B |
t |
p |
B |
t |
p |
B |
t |
p |
B |
t |
p |
||
|
Step 1 |
|||||||||||||
|
Sex |
0.02 |
0.55 |
0.581 |
-0.04 |
-1.12 |
0.265 |
-0.37 |
0.71 |
0.712 |
0.06 |
1.20 |
0.233 |
|
|
Age |
0.19 |
0.80 |
0.428 |
0.14 |
0.53 |
0.598 |
1.63 |
0.11 |
0.105 |
-0.52 |
-1.47 |
0.142 |
|
|
Class |
-0.17 |
-0.72 |
0.472 |
-0.11 |
-0.40 |
0.692 |
-1.25 |
0.21 |
0.214 |
0.40 |
1.13 |
0.259 |
|
|
Walking frequency |
0.02 |
0.45 |
0.654 |
-0.02 |
-0.62 |
0.533 |
-0.65 |
0.52 |
0.519 |
-0.11 |
-2.20 |
0.028 |
|
|
Walking time |
-0.04 |
-1.17 |
0.243 |
0.02 |
0.61 |
0.543 |
-0.02 |
0.99 |
0.988 |
-0.11 |
-2.14 |
0.033 |
|
|
Adjusted R2 |
0.017* |
0.028** |
0.047** |
0.042** |
|||||||||
|
Step 2 |
|||||||||||||
|
Social norm |
0.12 |
2.76 |
0.006 |
0.22 |
4.24 |
0.001 |
0.20 |
4.29 |
0.001 |
-0.139 |
-2.13 |
0.034 |
|
|
Safety attitude |
0.29 |
6.88 |
0.001 |
0.30 |
6.26 |
0.001 |
0.27 |
5.94 |
0.001 |
-0.024 |
-0.38 |
0.708 |
|
|
Family climate |
-0.63 |
-11.79 |
0.001 |
-0.37 |
-6.16 |
0.001 |
-0.44 |
-7.90 |
0.001 |
0.09 |
1.65 |
0.10 |
|
|
Adjusted R2 |
0.542** |
0.443** |
0.493** |
0.026** |
|||||||||
|
Step 3 |
|||||||||||||
|
SA* SN |
0.12 |
2.80 |
0.001 |
0.11 |
2.47 |
0.014 |
0.12 |
2.82 |
0.005 |
- |
- |
|
|
|
SA* FC |
-0.19 |
-4.88 |
0.001 |
-0.15 |
-3.32 |
0.001 |
-0.12 |
-2.86 |
0.004 |
- |
- |
|
|
|
SN* SA* FC |
0.176 |
3.30 |
0.001 |
- |
- |
|
- |
- |
|
- |
- |
|
|
|
Adjusted R2 |
0.046** |
0.028** |
0.023** |
|
|||||||||
|
Total R² |
0.605** |
0.499** |
0.563** |
0.068** |
|||||||||
Please review some methodological limitations such as social desirability (e.g. https://doi.org/10.1016/j.trf.2021.11.009).
Response: As the reviewer suggested, social desirability was added as a limitation. Please see page 12-13, 5. Implications and limitations. Or you can see the revision below.
.. A third limitation is that the relationships among the studied variables may be inaccurate due to social desirability. In the future, more objective indicators of crossing behaviours should be included
The manuscript is well written and considers variable factors. I would strongly encourage authors to also discuss traffic climate (e.g., https://doi.org/10.1016/j.aap.2018.01.031; https://doi.org/10.1016/j.trf.2025.01.030), as it may help explain other differences and cross-cultural variability for future studies.
Thanks very much for your valuable suggestion. Given the over length of the article, we decided not to added discussion about traffic climate in this article. The Traffic Climate Scale is mainly used in the field of driving. However, it could be an potential factor that attributed risky crossing behaviours of pedestrain aged 10-18 years. We are very interested in examining the validity of Traffic Climate Scale in crossing behaviours in the future.
